# Local and remote mean and extreme temperature response to regional aerosol emissions reductions

Daniel M. Westervelt[1,2], Nora R. Mascioli[3], Arlene M. Fiore[1,4], Andrew J. Conley[5], Jean-François Lamarque[5], Drew T. Shindell[6], Greg Faluvegi[2,7], Michael Previdi[1], Gustavo Correa[1], Larry W. Horowitz[8]

[1]Lamont-Doherty Earth Observatory, Columbia University. Palisades, New York, USA
[2]NASA Goddard Institute for Space Studies, New York, New York, USA
[3]University of California San Diego, School of Marine Sciences, San Diego, California, USA
[4]Department of Earth and Environmental Sciences, Columbia University, Palisades, New York, USA
[5]National Center for Atmospheric Research, Boulder. Colorado, USA
[6]Nicholas School of the Environment, Duke University. Durham, North Carolina, USA
[7]Center for Climate Systems Research, Columbia University, New York, NY, USA
[8]National Oceanic and Atmospheric Administration, Geophysical Fluid Dynamics Laboratory, Princeton, New Jersey, USA

*Correspondence to*: Daniel M. Westervelt (danielmw@ldeo.columbia.edu)

**Abstract.** The climatic implications of regional aerosol and precursor emissions reductions implemented to protect human health are poorly understood. We investigate the mean and extreme temperature response to regional changes in aerosol emissions using three coupled chemistry-climate models: NOAA GFDL-CM3, NCAR-CESM1, and NASA GISS-E2. Our approach contrasts a long present-day control simulation from each model (up to 400 years with perpetual year 2000 or 2005 emissions) with fourteen individual aerosol emissions perturbation simulations (160-240 years each). We perturb emissions of sulfur dioxide ($SO_2$) and/or carbonaceous aerosol within six world regions and assess the statistical significance of mean and extreme temperature responses relative to internal variability determined by the control simulation and across the models. In all models, the global mean surface temperature response (perturbation minus control) to $SO_2$ and/or carbonaceous aerosol is mostly positive (warming), statistically significant, and ranges from +0.17 K (Europe $SO_2$) to -0.06 K (US BC). The warming response to $SO_2$ reductions is strongest in the US and Europe perturbation simulations, both globally and regionally, with Arctic warming up to 1 K due to a removal of European anthropogenic $SO_2$ emissions alone; however, even emissions from regions remote to the Arctic, such as $SO_2$ from India, significantly warm the Arctic by up to 0.5 K. Arctic warming is the most robust response across each model and several aerosol emissions perturbations. The temperature response in the northern hemisphere mid-latitudes is most sensitive to emissions perturbations within that region. In the tropics, however, the temperature response to emissions perturbations is roughly the same in magnitude from emissions perturbations either within or outside of the tropics. We find that climate sensitivity to regional aerosol perturbations ranges from 0.5 to 1.0 K per W m$^{-2}$ depending on the region and aerosol composition, and is larger than the climate sensitivity to a doubling of $CO_2$ in two of three models. We update previous estimates of Regional Temperature Potential (RTP), a metric for estimating the regional temperature responses to a regional emissions perturbation that can facilitate assessment of climate impacts with integrated assessment models without requiring computationally demanding coupled climate model simulations. These calculations indicate a robust regional response to aerosol forcing within the northern hemisphere mid-latitudes, regardless of where the aerosol forcing is located longitudinally. We show that regional aerosol perturbations can significantly increase extreme

temperatures on the regional scale. Except in the Arctic in the summer, extreme temperature responses largely mirror mean temperature responses to regional aerosol perturbations through a shift of the temperature distributions and are mostly dominated by local rather than remote aerosol forcing.

## 1 Introduction

5        Understanding regional climate responses to present and future anthropogenic forcing agents remains a key challenge of direct relevance to human and natural systems. Emissions of aerosols and their precursors are spatially heterogeneous and short-lived, and thereby expected to exert complex responses as emissions of air pollutants are reduced through policies enacted to protect human health. Emissions of sulfur dioxide ($SO_2$), black carbon (BC), and organic carbon aerosol (OA) have decreased throughout the United States and Europe for several decades (Leibensperger et al., 2012; Tørseth et al., 2012). On

the other hand, emissions have largely increased in recent decades in countries such as China, India, and others in the Global South; however, since 2013, emissions of $SO_2$ are beginning to decline at least in China, while emissions in India continue to increase (Fontes et al., 2017; Li et al., 2017; Lu et al., 2011; Samset et al. 2019). As emissions of anthropogenic aerosols and their precursors are reduced in high-emitting regions such as China, their reduction is expected to perturb regional and global temperatures (Kasoar et al., 2016). To improve future climate projections, a deep understanding of the magnitude, spatial

pattern, statistical significance, and physical mechanisms of the temperature response to a phasing out of both scattering and absorbing anthropogenic aerosols is needed. Here we address this need by simulating the local and remote mean and extreme surface temperature responses to removal of different components of anthropogenic aerosols from six world regions in three distinct earth system models.

        The net effect of removal of global emissions of all anthropogenic aerosols is a surface warming, as decreases in

aerosol scattering result in additional solar energy reaching the surface of the Earth (Myhre et al., 2013). Removal or reduction of scattering aerosols on the regional scale will also result in surface warming on average. However, removal of global and regional emissions of black carbon or other absorbing aerosol is generally expected to induce a cooling at the surface, due to a net reduction in the absorption of incoming solar radiation (Bond et al., 2013; Ramanathan and Carmichael, 2008; Samset et al., 2018). In addition to influencing surface temperature directly by scattering or absorbing incoming solar radiation (aerosol

direct effect), aerosols also indirectly influence surface temperature by modulating cloud properties such as brightness and lifetime (aerosol indirect effects) (Albrecht, 1989; Twomey, 1977). Regional emissions perturbations of both scattering and absorbing aerosols also exert significant local and remote precipitation responses (Westervelt et al., 2017, 2018), though here we focus primarily on mean and extreme surface temperature responses.

        Several previous studies have considered the global and regional climate response to *global* reductions in aerosol and

precursor emissions using transient future simulations (e.g. Gillett and Von Salzen, 2013; Levy et al., 2013; Samset et al., 2018; Westervelt et al., 2015), finding a robust increase of up to about 1 K of surface warming by 2100 in response to decreasing aerosol burden. Recently, additional studies have quantified mean surface temperature responses and radiative

forcing to *regional* emissions changes of aerosol (Murphy, 2013). Kasoar et al. (2016) used three global climate models to estimate the global and regional surface temperature impacts from the removal of Chinese anthropogenic $SO_2$ emissions, finding hemispheric warming in two of the three models. Conley et al. (2018) also used three climate models to estimate the mean surface temperature response to a removal of $SO_2$ emissions from the United States alone, with warming over the United States and in the Arctic found to be as high as 0.5 K. Persad and Caldeira (2018) used the NCAR CAM5 (Community Atmosphere Model 5) to show that climate responses to identical aerosol emissions changes are significantly different depending on the region where emissions are perturbed. Using a different model and a different emissions perturbation format, Kasoar et al. (2018) find similar patterns of mean surface temperature response to aerosols from different regions. Both Kasoar et al. (2018) and Persad and Caldeira (2018) used a single model to estimate the temperature responses to regional anthropogenic aerosol emissions.

Reductions in regional aerosol emissions may also influence temperature extremes; however, the magnitude, statistical significance, and physical mechanisms of the greenhouse gas and aerosol impact on extreme events is also poorly understood (Horton et al., 2016). The Intergovernmental Panel on Climate Change Special Report on Managing the Risks of Extreme Events and Disasters to Advance Climate Change Adaptation (IPCC SREX; (IPCC, 2012)) identified forcing factors that are important on regional scales (such as aerosols) as a key challenge to further understanding of the anthropogenic causes of extreme temperature change. Recent studies have found a role for global aerosol reductions in heat waves (Zhao et al., 2019) and also in temperature extreme indices (Mascioli et al., 2016; Samset et al., 2018) as defined by Expert Team of Climate Change Detection and Indices (ETCCDI) (Sillmann et al., 2013). To our knowledge, the extreme temperature response to regional aerosol emissions reductions has not been previously studied.

In addition to understanding the changes in mean and extreme surface temperature response to aerosol reductions, it is vitally important to understand the effective radiative forcing (ERF) induced by aerosols, and how ERF relates to temperature response. ERF includes the instantaneous top of atmosphere radiative forcing plus rapid adjustments, i.e. the radiative impacts to the top of atmosphere energy budget which are not related to surface temperature. Radiative forcing exerted by anthropogenic aerosols is far more spatially inhomogeneous than that from well-mixed greenhouse gases, making generalization of the climate responses to anthropogenic aerosol emissions changes a more difficult task (Shindell, 2014). Additionally, radiative forcing in one region may result in different temperature response in local regions compared with remote regions. Shindell and Faluvegi (2009) began to address this by using an early version of the Goddard Institute for Space Studies ModelE chemistry-climate model to estimate temperature responses per unit radiative forcing for forcing perturbations in several wide latitude bands. Shindell (2012) also used these latitude bands to further develop the Regional Temperature Potential (RTP), a temperature response metric normalized by aerosol ERF to provide estimates regional temperature change. More recently, Lewinschal et al. (2019) used NorESM (Norwegian Earth System Model) to calculate similar metrics based on emissions. Simple climate metrics such as RTP coefficients can be used in the Integrated Assessment Modelling (IAM) and climate impacts community to rapidly and easily calculate the climate impact of different energy or climate mitigation policies without requiring computationally expensive coupled climate model simulations. Thus far, metrics such as RTP incorporated

into IAMs have been based on simulations with a single climate model. Future climate projections can benefit and improve from a multi-model approach that enables more robust estimates of mean and extreme regional surface temperature responses per unit radiative forcing from a given region.

The relationship between surface temperature response and associated ERF is not well understood for individual short-lived forcing agents such as regional aerosols. The climate sensitivity parameter, or the ratio between the temperature response to an external forcing and the forcing itself (K per W m$^{-2}$), is a widely used metric essential for projecting future climate change (Myhre et al, 2013; Marvel et al., 2016; Previdi et al., 2013). Estimation of equilibrium climate sensitivity (ECS) using coupled models has mostly occurred in the context of a doubling (or quadrupling) of $CO_2$ concentrations (2x$CO_2$) (Arrhenius, 1896; Callendar, 1938; Cox et al., 2018; Huber et al., 2014; Knutti et al., 2017; Knutti and Hegerl, 2008; Knutti and Rugenstein, 2015; Otto et al., 2013). A few studies estimating the ability of single forcing agents to change surface temperature (sometimes called "forcing efficacy") have found that anthropogenic aerosols have a greater forcing efficacy than $CO_2$ (Hansen, 2005; Marvel et al., 2016; Shindell, 2014). These findings, however, have come from single models using global reductions in aerosol and precursor emissions, despite substantial regional dependence and heterogeneity of aerosol forcing. Estimates of ECS based on modelling and modern and paleoclimatic observations should take into account the forcing efficacy of regional aerosol perturbations, which our approach can help inform.

We improve on past work by conducting an extensive set of computationally demanding simulations in three (instead of one) Coupled Model Intercomparison Project Phase 5 (CMIP5) era chemistry-climate models in which emissions of $SO_2$, BC, OA, and a combination of all three are set to zero or significantly reduced in one of six world regions (instead of latitude bands). Using these simulations, we estimate the local and remote regional surface temperature responses to reduced or removed aerosol and precursor emissions. We aggregate our results in each model to provide an estimate of robustness of the regional surface temperature response. In order to compare the surface temperature responses across models, regions, and forcing agents (including aerosols but also carbon dioxide), and to provide updated estimates of regional temperature response metrics as done in Shindell (2012), we estimate the climate sensitivity for a given region and forcing agent in each of our models, on a global and regional basis. We also report for the first time the extreme surface temperature response to regionally specific emissions reductions of aerosols and their precursors in three climate models.

## 2 Methods

### 2.1 Models and simulations

Our modeling framework has been previously described by Westervelt et al. (2018), Westervelt et al. (2017), and Conley et al. (2018). Briefly, we employ three coupled atmosphere-ocean-land-sea-ice climate models with fully interactive chemistry of aerosols and trace gases: 1) Geophysical Fluid Dynamics Laboratory Coupled Climate Model version 3 (GFDL-CM3) (Donner et al., 2011), 2) Goddard Institute for Space Studies ModelE2 (GISS-E2-R) (Schmidt et al., 2014), and 3) Community Earth System Model version 1 (CESM1) (Neale et al., 2012). The model configuration for each is very similar to that used for CMIP5. For further model description and model evaluation, we refer readers to Westervelt et al. (2017) and Naik

et al. (2013). Only CESM1 includes prognostic simulation of aerosol size distribution (Conley et al., 2018 and references therein). Of particular relevance for our results is the model treatment of black carbon. In GFDL-CM3, black carbon is internally mixed with only sulfate in the radiation code, whereas in CESM1, black carbon is internally mixed with all aerosol constituents within a given aerosol mode. In GISS-E2, black carbon is externally mixed with other aerosol species (Schmidt
et al., 2014).

We conduct for each model a long "present-day" control simulation of up to 400 years in length, forced by perpetual year 2000 (2005 for NCAR-CESM1) conditions, including all emissions of aerosols and their precursors and greenhouse gas concentrations. We also conduct individual regional aerosol perturbation simulations of at least 160 years and as long as 240 years in each model, in which the anthropogenic aerosol or aerosol precursor emissions for a certain region are completely
removed (100%) or reduced by the amount shown in Table 1. Aerosol emissions removals are instantaneous and we do not consider the effect of a long time-evolving drawdown. The first 20 years of the perturbation simulations are discarded in the response calculation. We choose the magnitude of relative emissions reductions in order to have roughly equivalent emissions decreases for a particular species across regions and models. As an example, "ISO2" refers to a simulation with perpetual year 2000 conditions (2005 for NCAR-CESM1), perturbed by setting all anthropogenic $SO_2$ emissions over India to zero. Other
than the regional aerosol emissions perturbation, all other model settings remain identical to the control. Long control and perturbation simulations allow us to establish statistical significance and separate forced responses from internal climate variability.

We also conduct a set of simulations for each perturbation and control in each model using modeled climatological fixed sea surface temperatures (SST) and sea ice cover (SIC) in order to calculate ERF. These simulations only use the
atmosphere and land components of the climate models and are not coupled to the ocean and sea ice models but are otherwise identical to our longer coupled model integrations. ERF is determined by differencing the perturbation simulation minus the control simulation. Estimates of ERF done in this manner include the instantaneous radiative forcing plus the rapid adjustments from the atmosphere and the land. For the aerosol perturbation simulations, the ERF is calculated based on 50 years of simulation data for CESM1, 80 years for GFDL-CM3, and 160 years for GISS-E2 (to allow detection of a smaller forcing
observed in that model). The ERF associated with a doubling of $CO_2$ ($2xCO_2$) is also calculated using the fixed-SST method from simulations similar to $4xCO_2$ fixed-SST simulations conducted for the CMIP5 experiments. For comparison, the present day minus pre-industrial aerosol ERF in CESM1, GFDL-CM3, and GISS-E2 is -1.52, -1.60, and -0.76 W $m^{-2}$, respectively (Allen et al., 2015). This version of the GISS-E2 model does not include the aerosol-cloud lifetime effect, resulting in a smaller ERF, as discussed below.

2.2 Statistical methods

We estimate the change in surface temperature between the control and perturbation simulations as the cotemporal annual mean differences (perturbation minus control), and perform a paired sample modified Student t-test where the pairs are cotemporal samples of the perturbation and the control. The modified t-test accounts for autocorrelation in the model surface

temperature time series by calculating an effective standard error, which utilizes an effective sample size based on the lag-1 autocorrrelation. A time series showing autocorrelation overestimates the number of independent samples when calculating statistical significance, but our approach, based on Conley et al. (2018) and Zwiers et al. (1995), corrects against this overestimation. We also use the False Discovery Rate procedure of Wilks (2016) on our t-tests over our gridded atmospheric data, which limits the fraction of erroneously rejected null hypotheses in a field of mutually correlated t-tests (at each grid point).

2.3 Extreme indices

To estimate extreme temperature responses to aerosol perturbations, we use the "FClimDex" Fortran package (http://etccdi.pacificclimate.org/software.shtml) developed by the Expert Team of Climate Change Detection and Indices (ETCCDI) to estimate 27 climate extreme indices. Daily minimum, maximum, and mean surface air temperature is input to the extremes package for each of our simulations for which daily data was available, including the control simulation. Cotemporal differences were then taken as for mean temperature, and we performed modified paired t-tests (perturbation and control) to assess significance. Extreme temperature analysis was not performed on all of our simulations, but rather a subset of simulations that demonstrated the highest mean temperature response. Further, we only perform extreme analysis on simulations conducted for at least 160 years of daily data, as shorter time periods are not sufficient to build up robust statistics. We discard the first 20 years of each perturbation simulation (as with the mean surface temperature analysis), and use the corresponding matching years in the control run when creating the differences. We focus our analysis on the TXx index, one of the most commonly analyzed extreme index in the existing literature. TXx is defined as the maximum of the maximum daily temperature in a given time period (e.g., over a model simulated year) (Sillmann et al., 2013). We explored results using other temperature indices and found the results to be qualitatively similar to the results for TXx, and thus do not include these additional indices in the main text (see Supplementary Information).

## 3 Global and regional mean surface temperature responses to regional aerosol emissions

### 3.1 Comparison across models

Figure 1 shows the ~160-240 year annual mean surface temperature response in each of the three models for 6 regional aerosol perturbations. An analogous figure for all of the remaining simulations can be found in the Supplementary Fig. S1. The change in temperature in Fig. 1 and all following figures is the "perturbation minus control", representing the temperature response to a removal or reduction of emissions of anthropogenic aerosols and their precursors. Generally, the response is overwhelmingly positive (warming) with large regions of statistical significance in each of the three models for most simulations. We find a larger temperature response in GFDL-CM3 and CESM1 (first and second columns of Fig. 1) compared to GISS-E2, consistent with to the smaller magnitude of aerosol ERF in GISS-E2 (see Sect. 5) resulting from a lack of a cloud lifetime effect in that model (Westervelt et al., 2017, 2018). In all three models, the largest remote temperature responses are

over the Arctic, owing to the well-established polar amplification phenomenon (Smith et al., 2019; Stjern et al., 2019). Surface temperature response is strongest in the US $SO_2$ and Europe $SO_2$ simulations in all three models, with annual mean local and remote temperature increases of up to 1 K or higher. Despite different regions of emissions perturbations, the salient features of the spatial distribution of surface temperature response are similar between the US $SO_2$, China $SO_2$, US ALL ($SO_2$, BC, and

OC combined), Europe $SO_2$, and EU ALL (Fig. S1) perturbations in all models, suggesting that aerosol forcing in northern hemisphere mid latitudes (NHML) induces a qualitatively consistent spatial response pattern. This pattern features strong Arctic warming, differential heating of the Northern Hemisphere compared to the Southern Hemisphere, strong local responses, and far-reaching remote responses across continents (e.g., European warming in response to US $SO_2$ emissions reductions). The response pattern is also similar to regional modifications of land surface albedo as reported in Seneviratne et

al. (2018). Climate responses to aerosol perturbations also can project onto known modes of climate variability, such as El Niño-Southern Oscillation (ENSO), as described in Westervelt et al. (2018). The temperature response to US $SO_2$ emissions removal in CESM1 (Fig. 1b) resembles an El Niño-like response, with cooling in the western tropical Pacific Ocean coupled with warming in the eastern tropical Pacific Ocean. In GFDL-CM3, most simulations regardless of region or aerosol species result in cooling (sometimes statistically significant) south of 60ºS along the Antarctic coast starting roughly at the 180º

meridian coupled with surrounding statistically significant warming (e.g. EU $SO_2$, Fig 1d), suggesting interaction with the Amundsen Sea Low (ASL), which exerts significant influence on Antarctic climate (Raphael et al., 2016). However, this is also a region of strong climate variability in GFDL-CM3.

Although the surface temperature response to Indian $SO_2$ and BC emissions reductions is small in all models, despite the tropical location of the emissions perturbation, changes in temperature still occur at both poles in all models, with some

statistical significance. Removal of black carbon emissions (Fig. 1p, q, and r) elicits a very different temperature response in each of the three models in spatial distribution, sign, and magnitude, indicating a strong dependence of the surface temperature response to different model assumptions for black carbon, including different mixing state assumptions. Additionally, as reported in Westervelt et al. (2018), aerosol ERF from India BC perturbations is small (ranging from -0.04 to 0.06 W m$^{-2}$ across the three models) and statistically insignificant, resulting in climate responses that may be influenced by internal

variability. The weak forcing in the black carbon simulations may also reflect the role of rapid adjustments (Stjern et al., 2017; Smith et al., 2018), including the semi-direct effect of BC on clouds (Allen et al., 2019). The climate response to BC perturbations in other regions, such as US BC (Fig S1 panel g and h), also is marked by disparate temperature responses, further highlighting the sensitivity of climate response to model physics, and in some cases representing noise when forcing signals are small. The role of transport of BC from source regions remote to the Arctic may also be a contributor to the Arctic

temperature response (Wang et al., 2014).

## 3.2 Robustness across models

To estimate robustness of the surface temperature responses to regional aerosol perturbations, we use the sign (warming or cooling) and the statistical significance as a point of comparison between the three models. Figure 2 shows the

agreement between models in sign and statistical significance in each of the aerosol perturbations simulations that were conducted by all three models. We find widespread agreement in sign and significance in the US $SO_2$ (Fig. 2a), Europe $SO_2$ (Fig. 2b), China $SO_2$ (Fig. 2c), and US All ($SO_2$, BC, and OA combined, Fig. 2d) simulations. Using sign agreement in three models as a minimum for a qualification of robustness (light blue color), the most robust responses are to Europe SO2 removal, where 81 % of the Earth's surface qualifies as robust (values in the upper right of Fig. 2 panels). On the other hand, the response to India BC is robust across only 39 % of the Earth's surface. We conclude that climate responses to black carbon over India exhibit large variability between models compared to climate responses from source regions such as the US and Europe, likely due to the small forcing exerted by the BC perturbation simulations.

The three models frequently agree in the sign and significance of Arctic warming, indicating that the Arctic surface temperature response is one of the most robust features of climate response to regional aerosol perturbations. Local responses are also robust, in particular the US $SO_2$ and US All perturbations show high levels of robustness (green and dark blue colors in Fig. 2a and 2d) over North America. The models agree in sign and significance in the remote Arctic temperature response even in the case of India BC and African biomass burning emissions perturbation, suggesting that the Arctic warming response is somewhat independent of emissions region or aerosol composition. Overall, all three models agree on sign and at least two report statistical significance over 32 % of the Earth's surface (66 % when not including significance) in response to removal of US $SO_2$ emissions.

### 3.3 Local and remote responses by region

In Fig. 3, we present the global and regional mean surface temperature response to 14 different emissions perturbations in each of the three models. The emissions reductions forcing these temperature changes are roughly the same across models within a given perturbation scenario (Table 1). The global mean surface temperature response (Fig. 3a) indicates warming in 33 of the 34 simulations (US BC in GFDL-CM3 being the only example of global cooling) and is significant at the 95% confidence level in 30 of the 34 perturbation simulations. The Europe and US emissions perturbations (e.g., ESO2, EALL, USO2, etc.) cause the largest global mean temperature increases across all regions and aerosol compositions, resulting in a global mean warming of about 0.15 K. The $SO_2$ perturbations tend to result in greater warming than OA or BC (which can also result in global cooling). CESM1 and GFDL-CM3 tend to warm more than GISS-E2, although not for all simulations.

We break down the regional climate response into latitude bands, following the approach used by Shindell and Faluvegi (2009), by regionally averaging the temperature responses from 60 to 90ºN (Arctic, Fig. 3b), 30 to 60ºN (Northern Hemisphere Mid Latitudes, NHML, Fig. 3c), 30ºS to 30ºN (tropics, Fig. 3d), and 30 ºS to 90ºS (Southern Hemisphere, Fig. 3e). Surface temperature increases approach 1 K regionally averaged over the Arctic (60 to 90ºN) in CESM1 and GFDL-CM3, with GISS-E2 simulating smaller but still often statistically significant warming responses. The Arctic responds most strongly to European aerosol perturbations (e.g. ESO2, EALL), perhaps owing to the greater proximity of the European continent to the Arctic region. However, even remote regional aerosol perturbations, such as India $SO_2$ (ISO2), or South American biomass burning (SABB) lead to Arctic warming in all of the models (Fig. 2), with some statistical significance. NHML temperature

changes (Fig. 3c) are mostly dominated by these local perturbations. On the other hand, the temperature response to the emissions perturbations local to the tropics (red labels in Fig. 3d) are roughly the same in magnitude and significance as the response to some of the "remote" perturbations. Emissions perturbations local to the tropics exert a larger temperature response in the Arctic than they do either locally or in the closer NHML region. In the Southern Hemisphere (Fig. 3e), we find consistent,

statistically significant warming in CESM1, but less warming in GFDL-CM3 and GISS-E2, owing to the localized Antarctic cooling in the case of GFDL-CM3. Overall, responses in the Southern Hemisphere are less statistically significant.

## 4. Extreme surface temperature responses to regional aerosol emissions

The response of temperature extremes (TXx, annual maximum of maximum daily temperature) averaged over the entire 160-240 simulation years is shown in Fig. 4 for each simulation in each model for which daily data were available. In

addition to the TXx extreme index, we have also analyzed a series of other indices, however the results are qualitatively similar so we only present TXx here (see Supplementary Information for additional indices). In general, we find increases in extreme temperature nearly everywhere both locally and remotely in most simulations, with a few exceptions such as the BC aerosol perturbations. Increases in extreme temperature are as large as 1 K especially near the source region of the particular perturbation simulation. Remote increases in extreme temperature are observed for several perturbations, for example

European $SO_2$ in NCAR-CESM1 and GFDL-CM3. Statistical significance is less abundant in GISS-E2, though we find increases of similar magnitude in GISS-E2 and the other two models. Over land, extreme temperature (TXx) can be equally or more sensitive to regional aerosol forcing than mean temperature, which can be seen by comparing temperature changes in Fig. 1 and Fig. 4. For example, TXx response to US $SO_2$ is mostly similar in magnitude or slightly larger than mean temperature over the eastern US in all three models. In contrast, mean temperature changes are strong (up to 1 K) over the Arctic, whereas

extreme temperature changes (TXx) are much smaller (< 0.3 K) and statistically insignificant. This is likely caused by the seasonality of Arctic amplification, which is a robust response to external forcing in every season except summer. TXx values mostly reflect summer temperature changes, when the maximum temperature throughout the year is likely to occur in the northern hemisphere. We confirm this by showing extreme temperature response for the winter months December, Janaury, and February (DJF, Fig. S2), in which Arctic extreme temperature responses are larger and consistent with mean temperature

responses. We conclude that the remote response relationship between mean and extreme temperatures is therefore strongly seasonally dependent.

Figure 5 shows the global (panel a) and latitude band averaged (panels b through e) extreme surface temperature response in each of the model simulations, analogous to Fig. 3 for mean surface temperatures. Another extreme temperature metric TX90p, or the percentage of days when the daily maximum temperature is greater than the 90th percentile, is shown in

Fig. S3 but is qualitatively similar to Figure 5. Global mean extreme surface temperature response is largest in GFDL-CM3 and CESM1 and in the Europe $SO_2$ (ESO2) and US $SO_2$ (USO2) simulations, in which the TXx response can approach about 0.2 K. Global mean TXx is only statistically significant for the ESO2 in GFDL-CM3 and CESM1, USO2 for CESM1, and ISO2 for GFDL-CM3. Changes in the extreme temperatures over the Arctic (Fig. 5b) are close to zero and statistically

insignificant, in contrast to Arctic mean temperature, which was heavily affected by many of the remote aerosol perturbations, though this is primarily caused by the seasonal dependence of Arctic amplification, as described above. TXx responses in the NHML (Fig. 5c) are dominated by local aerosol perturbations, reaching statistically significant increases of up to 0.4 K, while remote perturbations have no statistical significance. In the tropics and the Southern Hemisphere (Figs. 5d and 5e), there is almost no significant response in TXx to any aerosol perturbation. We conclude that although extreme temperature can be increased by remote aerosol perturbations in a few cases, in general the local forcing is a much greater control on extreme temperature, and remote responses are not nearly as large or significant for TXx compared to mean surface temperatures.

Figure 6 shows the eastern US and global mean surface temperature probability density function for each model for control and USO2. Each probability density function has been normalized such that the area under the curve is equal to unity. The bars represent the actual probability density for each temperature value, whereas the dashed curve is a fitted Gaussian kernel density estimation of the probably density. In each model both globally and regionally, there is a clear shift in the mean of the distribution, resulting in additional occurrence of temperature extremes. Each mean shift is also statistically significant at the 95% confidence level, except for the eastern US regional temperature distributions in GISS-E2. For the spatial average over the eastern US, the shape of the distributions remains unimodal and not skewed in GISS-E2 and GFDL-CM3, except for CESM1 which is not skewed in the control simulation but skewed in the perturbation. Global mean temperature distributions are consistently bimodal in the control and perturbation and generally not skewed. Overall, distribution shapes are mostly consistent, indicating that a mean shift is the statistical mechanism behind the increased temperature extremes.

## 5. Effective radiative forcing and climate sensitivity

### 5.1 Effective radiative forcing and surface temperature response

We use ~80 year fixed-SST and SIC atmosphere-only simulations in each of the three models to diagnose ERF due to each aerosol emissions perturbation. The global mean ERF from the 34 simulations ranges from about -0.1 to 0.3 W m$^{-2}$, though all but 6 simulations (several of the BC emissions perturbations) have ERF greater than zero. In Fig. 7, we plot global mean surface temperature response from the ~200 year coupled model simulations against global mean ERF for every perturbation simulation. We find a strong positive correlation among all models (r = 0.64 for CESM1, r = 0.79 for GFDL-CM3, and r = 0.76 for GISS-E2), consistent with previous studies (Liu et al., 2018; Marvel et al., 2016). There is substantial overlap and a similar slope for all three models (~0.4 K per W m$^{-2}$), indicating that, on a global mean basis, the models are each similarly sensitive to regional aerosol forcing. We further analyze the climate sensitivity to aerosol forcing in the following section.

### 5.2 Global climate sensitivity to regional aerosol perturbations and global CO$_2$ doubling

For a selection of simulations in which the aerosol ERF was statistically significant, we calculate in Fig. 8 the climate sensitivity parameter (K per W m$^{-2}$) to the regional aerosol perturbations as the quotient between the equilibrium global surface temperature response from the coupled model simulations and global ERF using the fixed-SST approach, similar to the

equilibrium climate sensitivity (ECS) approach used for $CO_2$. We also present the equilibrium climate sensitivity to a doubling of $CO_2$ ($2xCO_2$) in each of the three models using the same fixed-SST methodology for comparison to the aerosol climate sensitivity. We find that the climate sensitivity parameter for aerosol perturbations varies by model and by forcing, but mostly ranges from about 0.5 to 1.0 K per W m$^{-2}$ in each of the three models, which is comparable to the values for $CO_2$ sensitivity of approximately 1.0 K per W m$^{-2}$ in GFDL-CM3 and CESM1, and 0.5 K per W m$^{-2}$ in GISS-E2. Surface temperature appears to be most sensitive to European $SO_2$ emissions in GFDL-CM3, US $SO_2$ emissions in CESM1, and US ALL ($SO_2$, BC, and OA combined) emissions in GISS-E2. The $2xCO_2$ climate sensitivity and the aerosol climate sensitivity for the European $SO_2$, European ALL, and US $SO_2$ are approximately equivalent at about 1.0 K per W m$^{-2}$ for GFDL-CM3 and CESM1. The aerosol climate sensitivity is also in good agreement (overlapping error bars in Fig. 8) for the US $SO_2$ emissions perturbation between the three models. However, the aerosol climate sensitivity is often substantially greater than $2xCO_2$ climate sensitivity in GISS-E2, consistent with results from Marvel et al. (2016), discussed further below. Differences between aerosol climate sensitivity and $2xCO_2$ climate sensitivity can be explained by the differences in both the temperature response and the associated ERF for each perturbation. In particular, ERF may be quite different between heterogeneous, relatively smaller in magnitude forcing agents such as regional aerosols compared to more globally homogeneous, large forcing agents such as $CO_2$. Using 11 models including GISS-E2, Smith et al. (2018) found that rapid adjustments reduce the ERF for BC aerosol, but increase the ERF for $CO_2$ forcing, consistent with the hypothesis that differences in ERF can explain differences in the temperature sensitivities shown in Fig. 8.

Previous work by Marvel et al. (2016) and Hansen et al. (2005) using only the GISS-E2 climate model found that the forcing efficacy of global aerosol reductions is greater than that of $CO_2$. We extend this finding for GISS-E2 to regional aerosol emissions reductions, as the climate sensitivity parameter in all but one of our regional aerosol perturbation simulations in GISS-E2 is larger than the $2xCO_2$ perturbation. In contrast, the aerosol climate sensitivity parameter in both GFDL-CM3 and CESM1 is smaller or about equal to that of $2xCO_2$. We can conclude at minimum that aerosol forcing efficacy is model dependent, especially for regional aerosol perturbations, and this further highlights the importance of using multiple models to estimate or constrain estimates of ECS that includes forcing from a diverse set of agents. The CMIP6 experiments may be used to shed further light on the relative efficacy of aerosol and greenhouse gas forcing, though not from regional perturbations.

## 5.3 Regional Temperature Potential

In addition to the global temperature response and global ERF, we also estimate the regional temperature sensitivities. We use the approach of Shindell (2012), which introduced Regional Temperature Potential (RTP) coefficients. These coefficients account for the spatial heterogeneity of aerosol forcing and temperature response and can be derived for any pair of response regions and forcing regions. Following the methods of Shindell (2012) and Lewinschal et al. (2019), we calculate, within each latitude band, the temperature response to regional aerosol perturbations as a function of the latitude band averaged ERF containing each aerosol perturbation region. We then normalize this quantity by the global mean equilibrium temperature

response to global mean forcing, resulting in a dimensionless coefficient giving the equilibrium temperature response in latitude band $x$ to forcing in region $y$. The response latitude band $x$ can be any of the bands defined in Fig. 2, whereas forcing regions $y$ are the latitude bands containing each of our 14 regional aerosol perturbation locations, either 30-60ºN (NHML) or 30ºS-30ºN (tropics). As defined in Shindell (2012), RTP for a given pair of regions is:

$$RTP = \frac{\frac{dT_x}{dF_y}}{\frac{dT_{global}}{dF_{global}}}$$ (Equation 1)

where $dT$ is change in temperature and $dF$ is change in ERF. Because of the normalization by global mean temperature and global mean ERF, the RTP coefficients are unitless.

RTP coefficients in each latitude band for a given aerosol perturbation region are reported in Table 2 for GFDL-CM3, Table 3 for GISS-E2, and Table 4 for CESM1. We present only RTP values for which the corresponding ERF and temperature

response were statistically significant or for which data was available. The India, South America, and Africa entries in Tables 2-4 are based on a forcing average from the tropics since that region contains almost all of the statistically significant signal. All other values are based on NHML latitude band forcing average. Higher values of RTP indicate higher sensitivity of the particular response region to the aerosol forcing regions. RTP values from individual models provide a range of possible estimates. Figure 9 shows the multi-model mean RTP coefficients for a selection of regional aerosol perturbation simulations,

along with a mean of the NHML and tropics perturbations grouped together ("NHML tot.", "tropics tot."). Figure 9 indicates that the response to NHML forcing is consistent in all response regions regardless of where the aerosol forcing is longitudinally located within the NHML, as indicated by the similar RTP magnitudes in the first 4 clusters of bars (CSO2, ESO2, USO2, and UALL). Consistent with our earlier findings in Fig. 2, the Arctic always emerges as the most sensitive region to nonlocal aerosol forcing. After the Arctic, regional sensitivities are greatest for NHML, tropics, and Southern Hemisphere (SH) for

perturbations in the NHML (e.g. CSO2, ESO2, USO2, UALL). For tropical perturbations such as ISO2, SABB and AFBB, either the SH or the tropics are most sensitive, after the Arctic. Across each of the aerosol perturbations, the RTP coefficients are similar in magnitude when grouped by similar latitudinal forcing locations.

Our findings are similar to Shindell (2012), but we find a higher sensitivity in the Arctic to NHML forcing (1.49, Fig. 9 "NHML tot." versus RTP of 0.43 in Shindell (2012)). Shindell (2012) finds the Arctic is most sensitive to local forcing but

we lack a perturbation simulation to diagnose that response here. Shindell (2012) reported an Arctic RTP for tropical forcing of 0.36, close to that of NHML forcing, indicating that aerosol perturbations in the tropics is also important for Arctic climate response, which qualitatively agrees with our findings in Fig. 9. Averaging the RTP values corresponding to statistically significant ERF and temperature response within a single latitude band (for example, average RTP of USO2, ESO2, and CSO2) yields a close match with Shindell (2012) RTP values, especially in the NHML and tropics. Shindell (2012) report an RTP of

0.49 for NHML response to NHML forcing, very close to the average of our NHML forcings in Fig. 9, which is 0.46 (orange bar in Fig. 9 for "NHML tot."). The other response regions (tropics and southern hemisphere) compare moderately well with

Shindell (2012) for NHML forcing (0.25 versus 0.15 for the tropics and 0.1 versus 0.05 for the southern hemisphere). Shindell (2012) used an older model and an idealized forcing through an entire latitude band as opposed to our more realistic localized forcing, which may account for some of the differences in each region.

The uncertainty range in the final two clusters of bars in Fig. 9 give the range of RTP values for the total NHML forcing using the model individual values to construct a high and low estimate. For the NHML forcing cases, which include USO2, CSO2, and ESO2, the responses are robust across our models and there is little intermodel variation, as indicated by the small uncertainty range in each of the four response regions under "NHML tot.". For the tropical forcing cases, the models diverge (uncertainty bars under "Tropics tot." in Fig. 9), especially in the regions remote to the tropics. These results imply that the use of RTP coefficients or similar simple climate response metrics for remote responses to forcing in NHML regions are more robust and reliable than those for remote responses to forcing in tropical regions.

## 6. Summary and conclusions

Using three coupled chemistry-climate models, we conduct 160-240 year simulations in which aerosols of a specific type and from a specific region are set to zero (or greatly reduced) and compare to an otherwise identical control simulation in order to estimate the mean and extreme temperature response to regional aerosol emissions reductions. We estimate both the near-source local climate response and the remote response to regional aerosol emissions for both mean and extreme temperatures. Removal of regional aerosol emissions almost universally results in warming both globally and regionally, with some exceptions including perturbations of black carbon, an absorbing aerosol species. Surface warming is largest and most robust across models in response to $SO_2$ emissions reductions, particularly $SO_2$ from Europe and the US. Using a sign and significance approach to assessing robustness, we estimate that about 81% of the global surface area has a robust surface temperature response to European $SO_2$ reduction. All perturbations except for Indian BC have a spatial robustness of greater than 50%. Furthermore, the magnitudes of the responses are in agreement (overlapping ranges in globally and regionally averaged temperature responses in most perturbation simulations) in CESM1 and GFDL-CM3, but temperature changes are smaller in GISS-E2 due to weaker aerosol forcing. We find both local and remote statistically significant regional climate responses to regional aerosol emissions perturbations. Local emissions perturbations exert a strong warming response in the northern hemisphere mid latitude (NHML) regions including the US and Europe. Aerosol emission reductions from all world regions that we considered significantly increase mean temperature in the Arctic by up to 1 K (for emissions perturbations from Europe). Emissions reductions from the NHML exert a warming response in the tropics that rivals the magnitude of the response to emissions perturbations that are local to the tropics.

We assess the climate sensitivity to aerosol perturbations in each model and find a range from about 0.5 to 1.0 K per W m$^{-2}$. The aerosol climate sensitivity varies by type of forcing (e.g. $SO_2$, OC, BC) and also magnitude of forcing, and can be

different than the 2xCO$_2$ climate sensitivity, due to differences between a heterogeneous, localized aerosol forcing and a more homogeneous CO$_2$ forcing. Though it has been argued that uncertainty in aerosol forcing is the major factor in uncertainty of estimates of climate sensitivity to CO$_2$ based on modern observations (Andreae et al., 2005), less attention has been given to the temperature sensitivity to aerosol forcing itself, both in response to global and regional aerosol perturbations. In contrast

to previous findings using global aerosol reductions (Hansen, 2005; Marvel et al., 2016), we find that the climate sensitivity to aerosol forcing is less than or equal to the climate sensitivity to a doubling of CO$_2$ in 2 of 3 models, indicating a strong dependence on both model choice and region of aerosol reduction. Future work using the CMIP6 simulations may shed light on forcing efficacy of global aerosol reductions using a large number of models.

We estimate updated RTP coefficients in order to help facilitate estimation of climate impacts metrics at a sub-global

scale. These updated RTP coefficients may be useful for integrated assessment modelling (IAM), such as the Long-range Energy Alternatives Planning system – Integrated Benefits Calculator (LEAP-IBC) (Heaps 2016), to calculate climate impacts across a range of emission scenarios quickly and efficiently. We improve on previous studies by providing RTP coefficients for multiple models and for a large variety of aerosol types and regional perturbations, and by narrowing the forcing region from latitudinal bands to specific countries or continents (e.g. US SO$_2$, European SO$_2$). We provide a multimodel mean RTP

as well as the range represented by individual models. We find that the regional temperature response to northern hemisphere mid-latitude forcing (NHML) is largely independent of longitudinal forcing location within the NHML. We also find a small range of intermodel variability in regional temperature response to NHML forcing, indicating robustness of the RTP coefficients. For aerosol forcing occurring in the NHML, our reported RTP coefficients are similar to those reported in Shindell (2012), except for the response in the Arctic, which we find to be more sensitive to NHML forcing. Our results indicate that

RTP coefficients for Arctic response to aerosol forcing in the Arctic may need to be revised upwards, which has implications for climate impacts and integrated assessment modelling applications. Further unexpected warming in the Arctic from the unmasking of aerosol forcing could bring about Arctic climate tipping points such as permafrost thawing even sooner than currently projected. Future work will link climate responses directly to emissions changes for each of our models, similar to what has been done for NorESM in Lewinschal et al. (2019).

We also consider the extreme temperature response to regional aerosol perturbations and find that by shifting the overall surface temperature distribution, aerosol perturbations increase the warming extremes (upper tail of the surface temperature distribution). The annual maximum of maximum daily temperatures, or TXx, increases by about 0.1 to 0.2 K globally, closely mirroring the global changes in mean surface temperature, suggesting a mean shift of the temperature distribution to warmer temperatures, with limited impact on the shape of the distribution mainly occurring in only one of our

models. We find the mean shift to be statistically significant on a global mean basis in all models, and regionally in two of the three models. Compared to mean surface temperatures, extreme temperatures are not very sensitive to remote aerosol perturbations, with a few exceptions.

The understanding of the major drivers of projected regional climate change is key information needed by the climate assessment and impacts community. Our results have the potential to provide a framework for a key methodological link

between physical science and impacts, adaptation, and vulnerability analysis. This work is a first step towards providing statistical relationships between the changes in regional aerosol emissions and the statistically significant changes in climate that can be attributed to them. Such relationships would allow for the generation of regional climate change scenarios without having to simulate computationally demanding chemistry-climate models.

## 5 Author Contributions

DMW wrote the manuscript and created all figures. NRM performed ERF simulations and contributed extremes analysis. AMF, DTS, and JFL originally conceived the project, with later input from DMW, AJC, MP, and LWH. GF, AJC, and GC conducted simulations and transferred data. All authors contributed to editing the manuscript.

## Code Availability

10    The code for the atmospheric component of the GFDL-CM3 model is available here: https://www.gfdl.noaa.gov/am3/. NCAR-CESM1 model code is available here: http://www.cesm.ucar.edu/models/cesm1.0/. GISS-E2 model code is available here: https://simplex.giss.nasa.gov/snapshots/.

## Data Availability

RTP coefficients have been provided here: https://figshare.com/articles/RTP_coefficients_Westervelt_et_al_ACP/10669322
15 (Westervelt, 2019). Global and regional temperature response model data used in the figures is provided here: https://figshare.com/articles/Global_mean_T_by_latitude_band/10710722 (Westervelt, 2019). Contact the corresponding author for any other data requests.

## Competing Interests

The authors declare no competing interests.

## 20 Acknowledgements

Funding for this study was provided by an NSF EaSM-3 grant AGS 14-19398. The authors declare no conflicts of interest, and views, opinions, and findings presented in this paper are solely those of the authors and do not reflect the views of the funding agency. The NCAR-CESM work is supported by the National Science Foundation and the Office of Science (BER) of the U.S. Department of Energy. NCAR is sponsored by the National Science Foundation. GISS-E2-R simulations 25 used resources provided by the NASA High-End Computing (HEC) Program through the NASA Center for Climate Simulation (NCCS) at Goddard Space Flight Center. We acknowledge Dr. Claudia Tebaldi for useful discussions on statistical methods and temperature extremes.

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

**Table 1:** Simulation description and labels, and amount of emissions perturbation (roughly the same for each model) in absolute terms and the percentage removed

| Simulation name | Region of emissions perturbation | Species perturbed | Perturbation amount (Tg yr$^{-1}$), (%) |
|---|---|---|---|
| ESO2 | Europe | Sulfur dioxide | 18 (80%) |
| EBC | Europe | Black carbon | 0.8 (100%) |
| EOC | Europe | Organic carbon | 2 (100%) |
| EALL | Europe | Sulfur dioxide, black carbon, organic carbon | (Sum of above) |
| USO2 | United States | Sulfur dioxide | 15 (100%) |
| UBC | United States | Black carbon | 0.4 (100%) |
| UOC | United States | Organic carbon | 0.8 (100%) |
| UALL | Unites States | Sulfur dioxide | (Sum of above) |
| CSO2 | China | Sulfur dioxide | 15 (80%) |
| ISO2 | India | Sulfur dioxide | 5.6 (100%) |
| IBC | India | Black carbon | 0.6 (100%) |
| IOC | India | Organic carbon | 2.78 (100%) |
| SABB | South America | Biomass burning sulfur dioxide, black carbon, organic carbon | 0.4 (SO$_2$), 0.4 (BC), 4.7 (OA) (100%) |
| AFBB | Africa | Biomass burning sulfur dioxide, black carbon, organic carbon | 0.4 (SO$_2$), 0.4 (BC), 5.3 (OA) (33%) |

**Table 2:** Regional Temperature Potential (RTP) values for GFDL-CM3 for simulations with statistically significant ERF and temperature response

| | 60-90ºN (Arctic) | 30-60ºN (NHML) | 30ºS-30ºN (Tropics) | 30-90ºS (SH) |
|---|---|---|---|---|
| CSO2 | 1.86 | 0.54 | 0.44 | 0.05 |
| ESO2 | 2.26 | 0.58 | 0.24 | 0.12 |
| EALL | 1.29 | 0.38 | 0.18 | 0.16 |
| USO2 | 1.43 | 0.42 | 0.21 | 0.09 |
| UALL | 1.87 | 0.32 | 0.22 | 0.11 |
| ISO2 | 2.98 | 0.45 | 0.50 | 0.41 |
| SABB | 4.57 | 0.36 | 1.21 | 0.44 |
| AFBB | 2.15 | 0.34 | 0.26 | 0.17 |

**Table 3:** Regional Temperature Potential (RTP) values for GISS-E2 for simulations with statistically significant ERF and temperature response

|        | 60-90ºN (Arctic) | 30-60ºN (NHML) | 30ºS-30ºN (Tropics) | 30-90ºS (SH) |
|--------|------------------|----------------|----------------------|--------------|
| CSO2   | 1.34             | 0.34           | 0.16                 | 0.22         |
| ESO2   | 0.62             | 0.43           | 0.23                 | 0.12         |
| USO2   | 0.87             | 0.37           | 0.16                 | 0.12         |
| UALL   | 0.80             | 0.44           | 0.15                 | 0.04         |
| SABB   | 0.97             | 0.61           | 0.42                 | 0.37         |
| AFBB   | 0.98             | 0.31           | 0.44                 | 0.45         |

**Table 4:** Regional Temperature Potential (RTP) values for CESM1 for simulations with statistically significant ERF and temperature response

|        | 60-90ºN (Arctic) | 30-60ºN (NHML) | 30ºS-30ºN (Tropics) | 30-90ºS (SH) |
|--------|------------------|----------------|----------------------|--------------|
| USO2   | 2.02             | 0.57           | 0.20                 | 0.52         |
| AFBB   | 1.29             | 0.72           | 0.97                 | 2.37         |

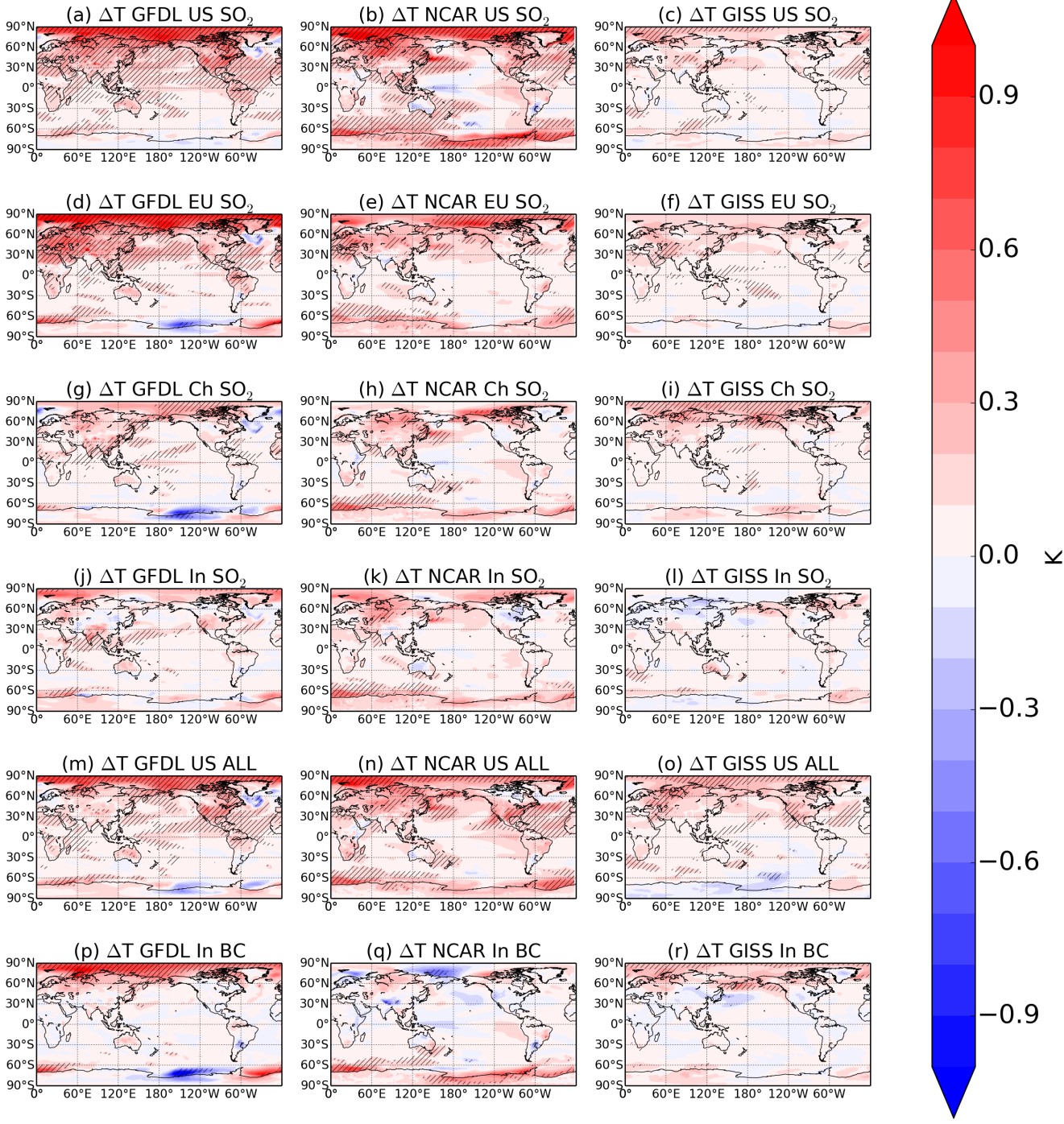

Figure 1: 200-year annual mean temperature responses (K) to aerosol emissions decreases in each of the three models (GFDL-CM3, first column; NCAR-CESM1, second column; GISS-E2, third column) for several different regional emissions decreases

(simulations indicated in figure titles; see Table 1). Hatching represents statistical significance at the 95% level according to a Student's t-test with the False Discovery Rate method from Wilks (2016) applied.

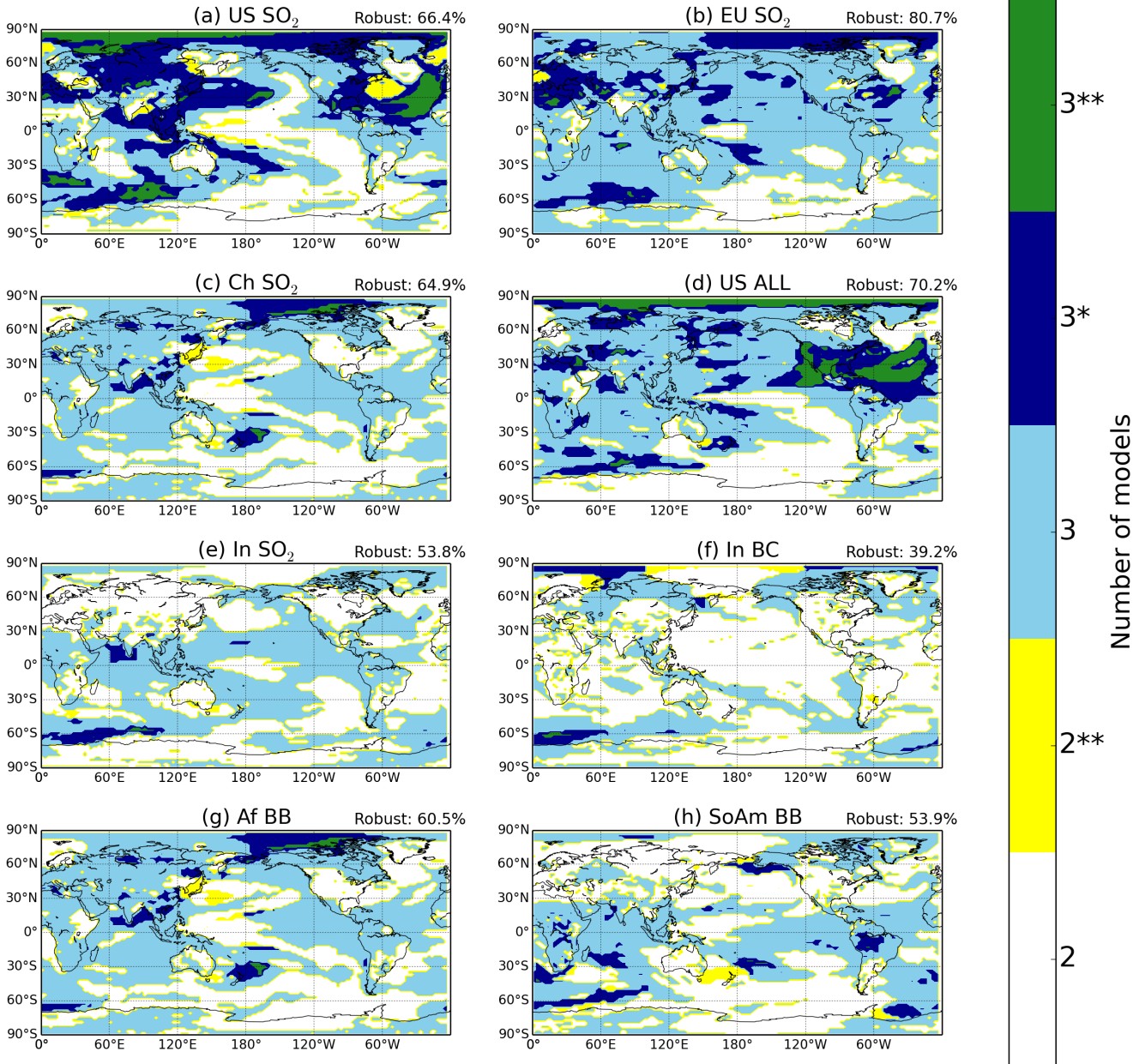

Figure 2: Regions of robustness in surface temperature response to individual aerosol emissions perturbations (panels a-h). The different colors represent the number of models in agreement in sign (two or three) for a particular location, and asterisks indicate whether models agree that the response is statistically significant (** for significance in all three or both models, * for significance in two out of three models, and no asterisks for significance in 1 or no models). Robustness indicates percentage of the surface area that has all 3 models in sign agreement.

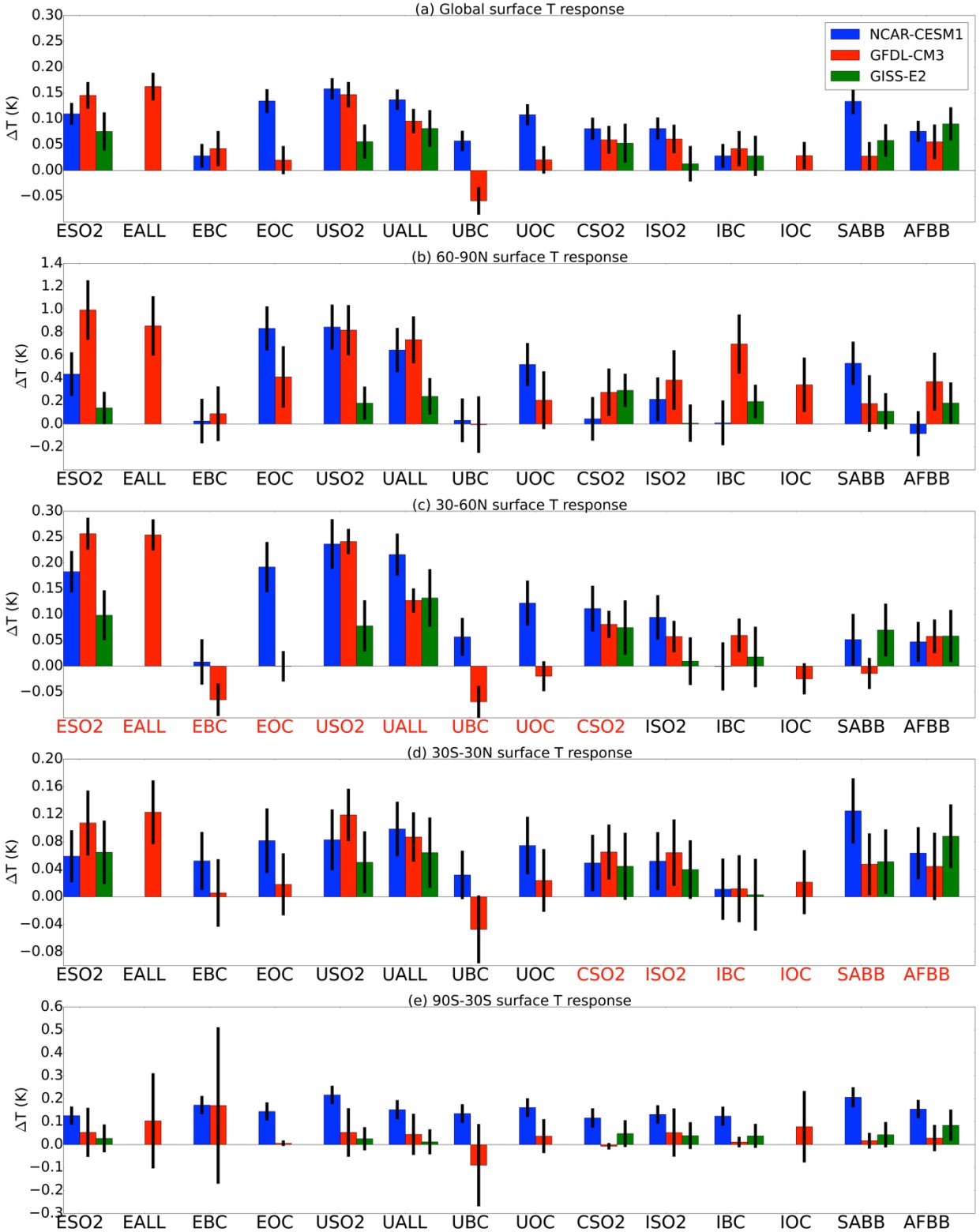

Figure 3: Global annual mean (panel a) and regional mean by latitude bands (panels b through e) surface temperature responses (K) to each of the 14 aerosol perturbation simulations. Error bars show ±2 standard errors of the mean to assess statistical significance. Regions that are "local" to the given latitude band are in red. See Table 1 for definition of abbreviations. Note the different scales in each panel.

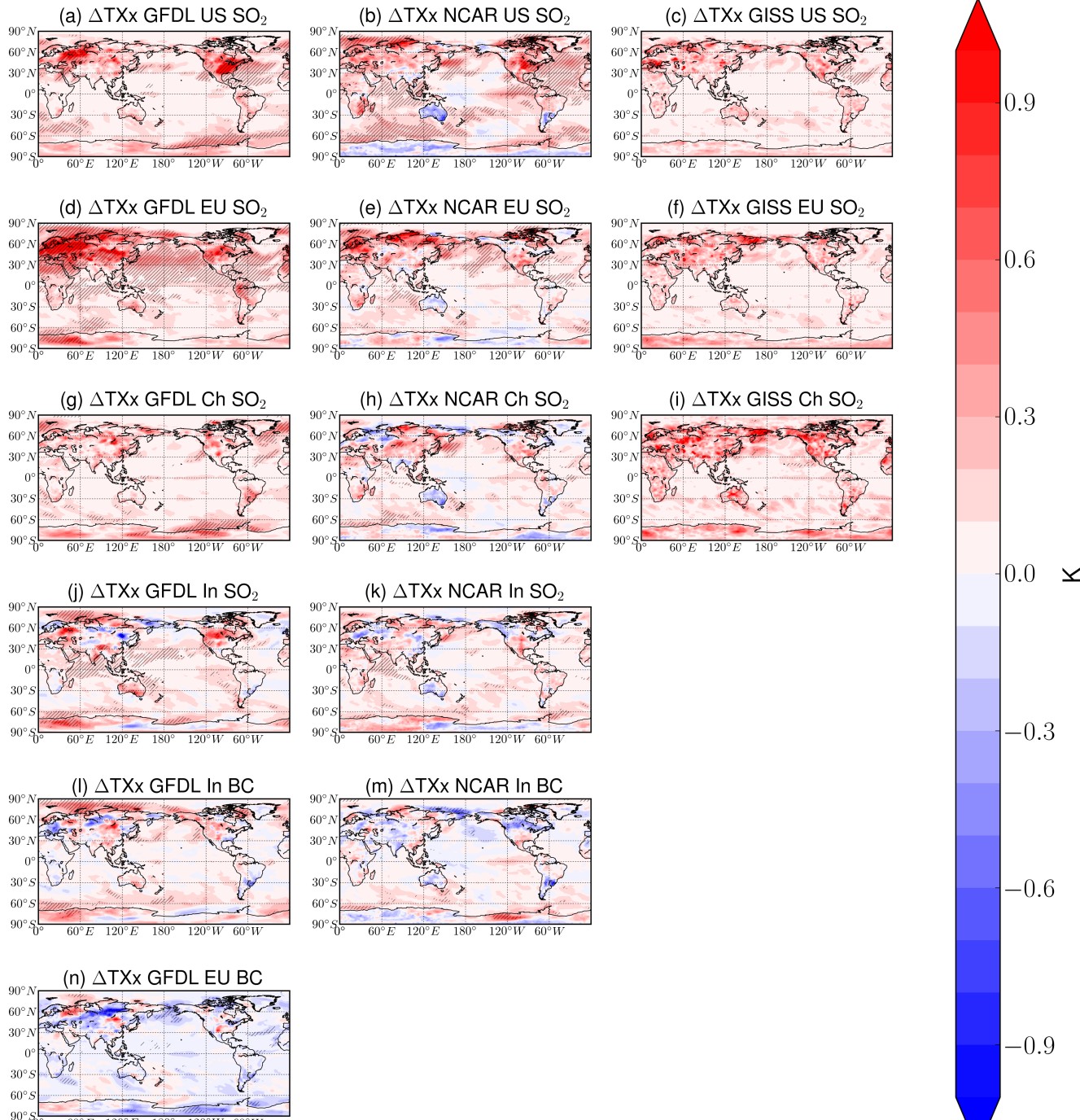

Figure 4: 200-year annual extreme temperature (TXx) responses (K) to aerosol emissions decreases in each of the three models (GFDL-CM3, first column; NCAR-CESM1, second column; GISS-E2, third column) for several different regional emissions decreases (simulations indicated in figure titles; see Table 1). Hatching represents statistical significance at the 95% level according to a Student's t-test with the False Discovery Rate method from Wilks (2016) applied.

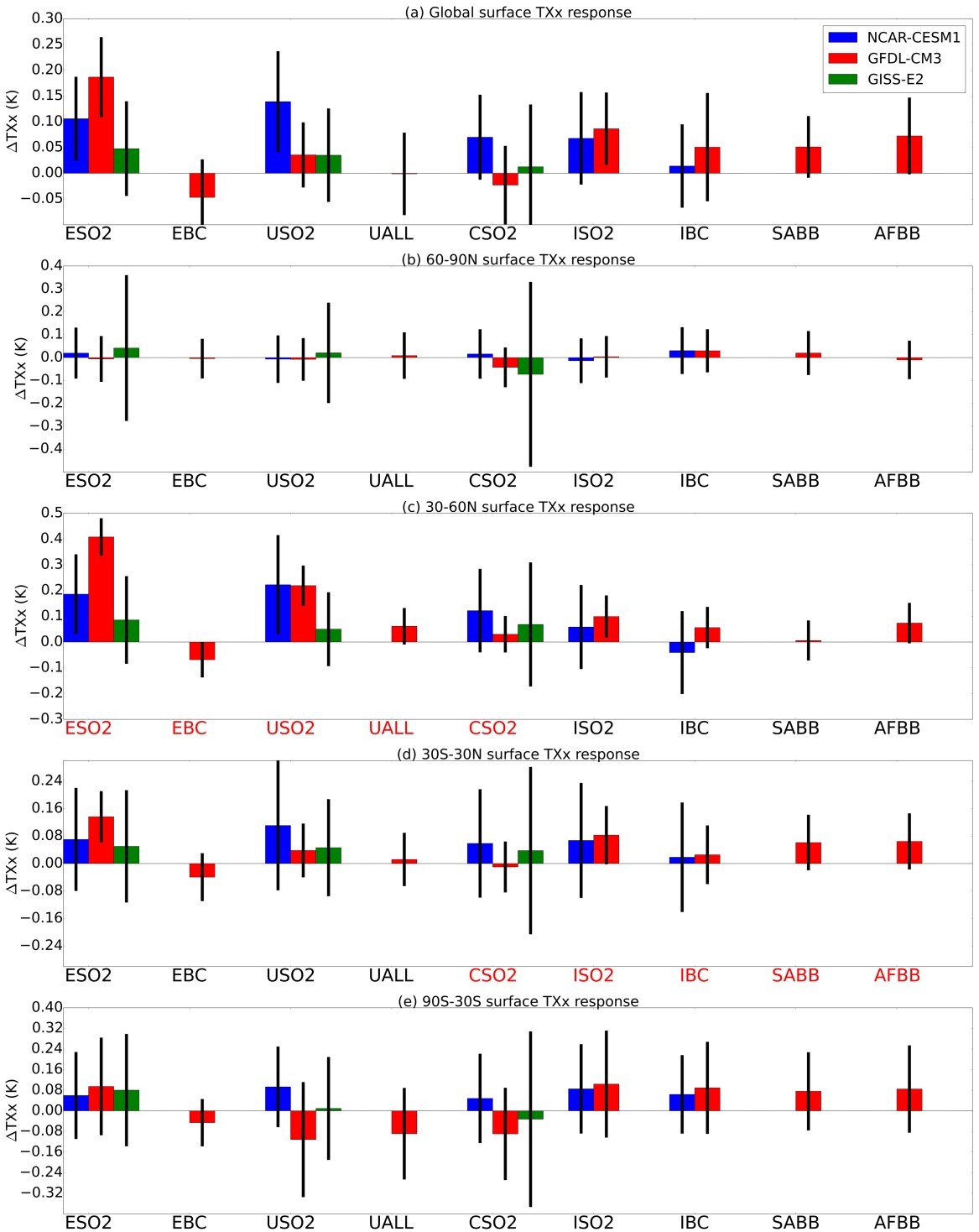

Figure 5: Global annual mean (panel a) and regional mean by latitude bands (panels b through e) extreme temperature responses (K) to each of the 14 aerosol perturbation simulations. Error bars show ±2 standard errors of the mean to assess statistical significance. See Table 1 for definition of abbreviations

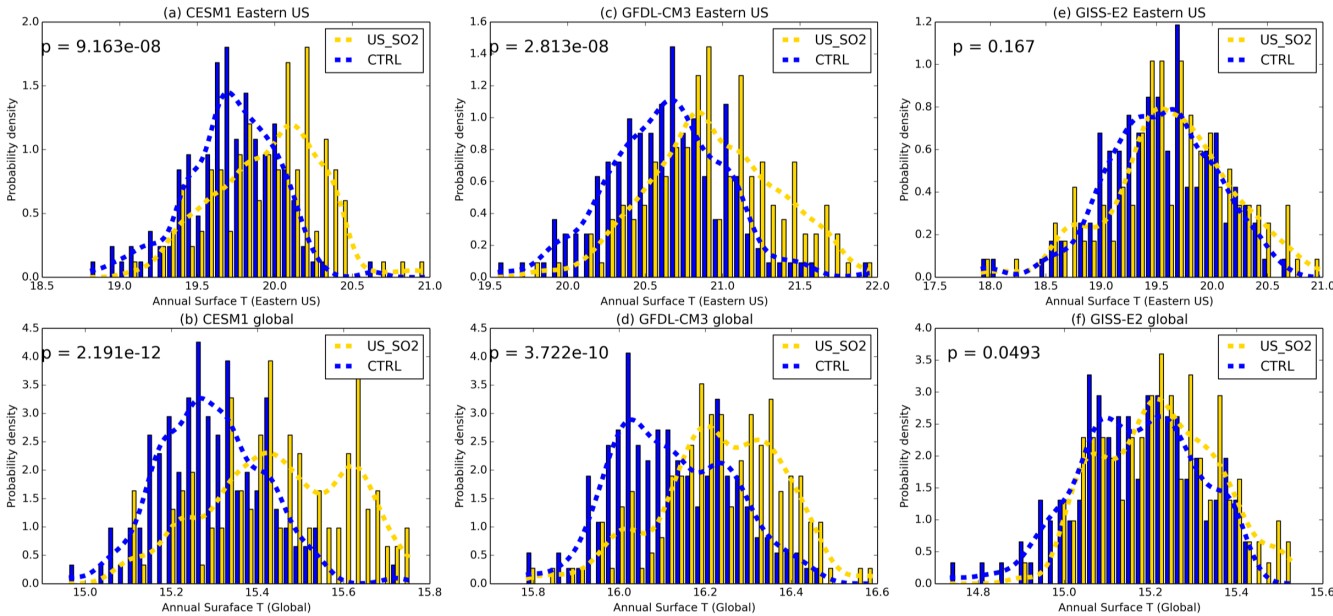

Figure 6: Eastern US regional (top row) and global mean (bottom row) probability density function for control and perturbation simulations in each model (columns). Dashed line is the Gaussian kernel density estimation for the normalized probability density function.

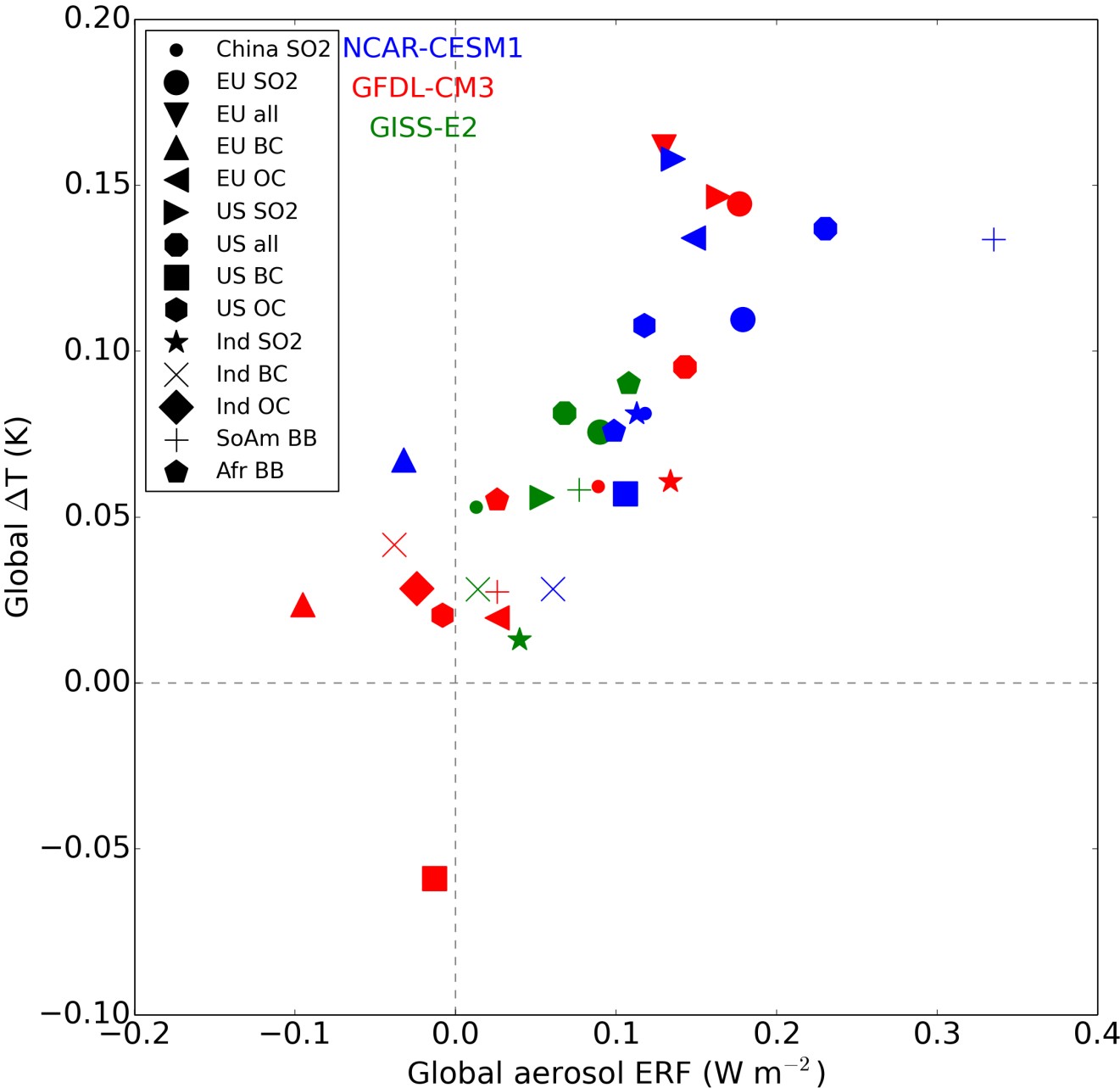

Figure 7: Scatterplot of global mean surface temperature response (K) to regional aerosol perturbations (symbols) versus global mean effective radiative forcing in each model (green: GISS-E2, red: GFDL-CM3, blue: NCAR-CESM1).

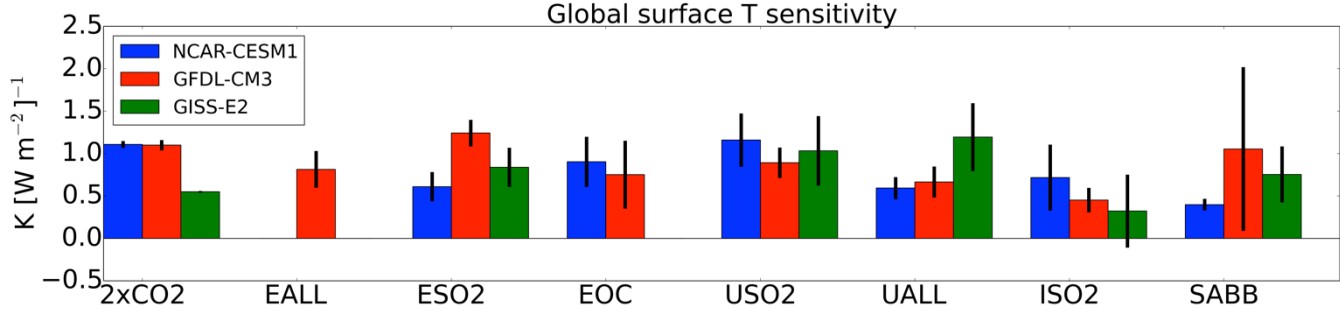

Figure 8: Global climate sensitivity to regional aerosol emissions perturbations and to a doubling of $CO_2$ (2x$CO_2$) in each model. Error bars represent ±2 standard errors around the mean. See Table 1 for definition of abbreviations.

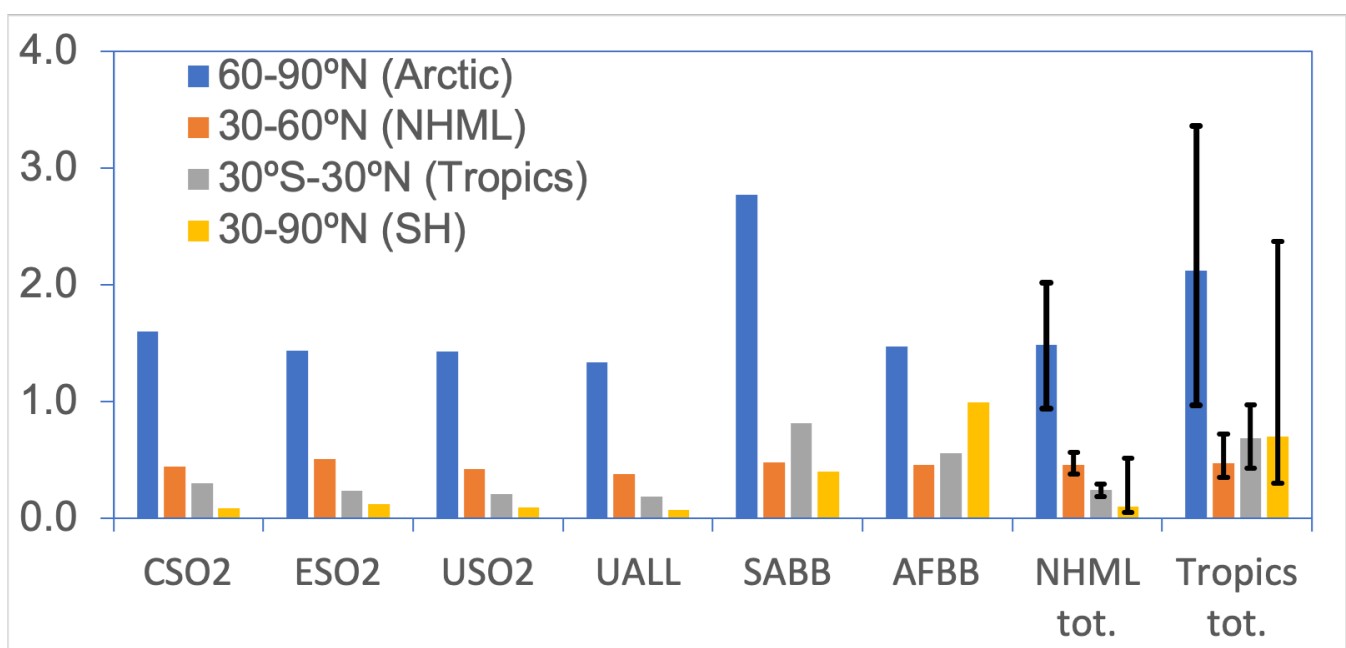

Figure 9: Regional Temperature Potential (RTP) coefficients (unitless) for the multi-model mean between GFDL-CM3, GISS-E2, and CESM1 for select simulations and the average by forcing region (e.g. "NHML tot." and "Tropics tot."). Uncertainty bars in the last two columns indicate the range of the RTP values as reported by the three models.

