# Peer review of "Local and remote mean and extreme temperature response to regional aerosol emissions reductions"

_Atmospheric Chemistry and Physics, 2019_

## Referee Comment (RC1) · Anonymous Referee #1 · 27 Dec 2019

This is a nice study. It's great to see some numbers on these effects, especially in the context of a multi-model study. This paper is clearly important, not only for its additions to the scientific literature, but also for policy at the intersection of climate and air quality, as well as for negotiations of allowable greenhouse gas emissions. The paper is also quite well written. I am recommending minor revisions.

Comments:

I really appreciate your careful attention to your statistical tests, particularly the number of degrees of freedom.

GISS ModelE has a configuration (TOMAS) that could allow you to prognostically sim-

ulate aerosol size distribution, like CESM1. It would be useful to say why you chose not to use that configuration.

In your introduction, a reference to Murphy (2013) seems relevant: https://www.nature.com/articles/ngeo1740

Section 2.3: It would be helpful to spell things out a little more when computing the ETCCDI metrics. For your baseline, did you use the entire 400-year control run? Did you throw out any of that period for spinup? For the perturbed case, did you throw out any of the 160 years or did you use the whole thing? (This sort of thing really matters for TX90p, but to some degree for the others as well.)

Page 6, lines 24-25: In light of Malavelle et al. (2018) (https://www.nature.com/articles/nature22974), it might be worth making a comment as to whether the cloud lifetime effect is actually something that should be included.

Page 6, lines 24-29: Your results bear a striking similarity to those of Seneviratne et al. (2018): https://www.nature.com/articles/s41561-017-0057-5 Your choice, but could be worth a comment?

Page 10, lines 1-3: Is there a statistical test you could use? Anything better than simply eyeballing the change?

Figure 1: The results here say some interesting things about aerosol transport. In particular, the bottom row indicates that Wang et al. (2014) (https://agupubs.onlinelibrary.wiley.com/doi/full/10.1002/2014JD022297) and subsequent related studies might be relevant. The point being, your discussion about common patterns of temperature change is good, but it's not necessarily the entire story.

Figure 2: I'd recommend getting rid of the red color and just leaving it white. It doesn't say much that two models agree on the sign but without statistical significance.

In Figure 4, CESM1 clearly has some different Southern Hemisphere behavior from

the other two models (which doesn't show up in Figure 5, possibly due to cancellation). Any hypotheses as to what might be going on here?

---

## Referee Comment (RC2) · Anonymous Referee #2 · 9 Jan 2020

Review of "Local and remote mean and extreme temperature response to regional aerosol emissions reductions" by Westervelt at al.

Summary This paper quantifies the temperature response, including extremes, due to removal of aerosol emissions in six different geographical regions, with three different climate models. Such removal generally leads to warming, with the Arctic region particularly vulnerable. Similar changes occur in temperature extremes. The authors also update Regional Temperature Potentials, which are useful for estimating regional climate impacts due to regional aerosol changes in Integrated Assessment Models.

Overall, the paper is well written and contains interesting and novel results. I recom-

mend publication with minor revision.

Comments P2 L12. See also "Emerging Asian aerosol patterns" Nature Geosci. 12 (2019) by Samset et al.

There are also several PDRMIP papers that are relevant to the discussion, that should probably be cited. For example, "Rapid Adjustments Cause Weak Surface Temperature Response to Increased Black Carbon Concentrations" by Stjern et al. JGR 2017.

Also, "Arctic Amplification Response to Individual Climate Drivers" by Stjern et al. JGR 2019. Looks like the Arctic Amplification paper is eventually mentioned on Page 6.

P4 L33. Naik reference parentheses typo.

Model Description Section. Please include some brief information of each of these model's aerosol ERF (e.g., PD-PI). For example, in "A 21st century northward tropical precipitation shift caused by future anthropogenic aerosol reductions" by Allen JGR 2015, CESM yields - 1.52; GFDL yields -1.60; and GISS yields -0.76 W/m^2. So GISS has a much smaller aerosol ERF. Based on this alone, I would expect any temperature response to similar aerosol perturbations to be weakest in GISS. You may also want to mention why this is. For example, GISS lacks aerosol-cloud second indirect effects (aerosol-cloud lifetime effect). Looks like this is eventually mentioned on Page 6. Might be beneficial to mention here as well.

Although, it is interesting that GISS also has a smaller climate sensitivity to CO2, relative to GFDL and NCAR. This could also contribute to a weaker surface temperature response in GISS.

Methods Section. These simulations are designed for an instantaneous removal of aerosol emissions. Thus, there is no time evolution of the aerosol emission reductions, which would be more realistic. How does this impact the results? Maybe a few comments on this are necessary.

Methods Section. It is stated perturbation simulations are conducted for 160-200 years.

What years are actually analyzed? I assume the first several decades (hopefully more) are not used, since the model is not yet in some sort of quasi-equilibrium?

P6 L26. "Surface temperature response is strongest in the US SO2 and Europe SO2 simulations in all three models, with annual mean local and remote temperature increases of up to 1 K or higher." I assume this is related to the magnitude of the aerosol emission reduction. What if you normalize by this perturbation? I guess you eventually normalize by ERF, which would be similar.

P7 L10. Regarding the weak and inconsistent BC temperature response. Again, this appears to be consistent with the Stjern PDRMIP paper above.

Also, this may be related the resulting rapid adjustments. See, for example, Smith, C. J. et al. "Understanding rapid adjustments to diverse forcing agents". Geophys. Res. Lett. 45, 12023–12031 (2018).

And in particular, the impact of the vertical profile of absorbing aerosol on the rapid adjustments. For example, Allen, R. J. et al. "Observationally constrained aerosol cloud semi-direct effects" npj Climate and Atmospheric Science (2019) showed very different surface temperature responses to absorbing aerosol dependent on the vertical absorbing aerosol profile. Models tend to have a vertically-uniform profile, and this leads to a negative adjustment, associated with high cloud reductions. However, a vertical profile that resembles CALIPSO observations (more absorption in the lower troposphere) yields low/mid-level cloud reductions, a positive adjustment, and surface warming. Looks as if this is eventually discussed near P11.

Figure 2. Can the "Robustness %" be defined in the figure caption.

Near Page 8 L10. Can you explain why BC reductions lead to surface warming (despite having a positive RF) in nearly all cases?

How do tropical emission reductions drive Arctic warming? Is it due to direct transport of aerosol to the Arctic, or due to changes in atmospheric/oceanic circulation that

then leads to Arctic warming? Are these responses robust across the three models? Perhaps this is outside the scope of the current study.

---

## Author Comment (AC1) · 17 Jan 2020

**Response to Referees for Westervelt et al. (2020):**

**Reviewer 1**

This is a nice study. It's great to see some numbers on these effects, especially in the context of a multi-model study. This paper is clearly important, not only for its additions to the scientific literature, but also for policy at the intersection of climate and air quality, as well as for negotiations of allowable greenhouse gas emissions. The paper is also quite well written. I am recommending minor revisions.

We thank the reviewer for the nice comments and suggestions.

Comments: I really appreciate your careful attention to your statistical tests, particularly the number of degrees of freedom.

GISS ModelE has a configuration (TOMAS) that could allow you to prognostically simulate aerosol size distribution, like CESM1. It would be useful to say why you chose not to use that configuration.

TOMAS introduces an extra ~200 or so advected tracers for aerosol size, so running 200 to 400 year simulations with it is currently not computationally efficient. It is worth noting that the first author (Westervelt) is a TOMAS developer and did his PhD on TOMAS.

In your introduction, a reference to Murphy (2013) seems relevant: https://www.nature.com/articles/ngeo1740

Good point, added to manuscript. The following sentence has been edited to include radiatve forcing and a citation was added for Murphy (2013):

"Recently, additional studies have quantified mean surface temperature responses and radiative forcing to *regional* emissions changes of aerosol (Murphy, 2013)."

Section 2.3: It would be helpful to spell things out a little more when computing the ETCCDI metrics. For your baseline, did you use the entire 400-year control run? Did you throw out any of that period for spinup? For the perturbed case, did you throw out any of the 160 years or did you use the whole thing? (This sort of thing really matters for TX90p, but to some degree for the others as well.)

This is a good point. We remove about the first 200 years or so of the control runs and treat that as spin up. Then we also remove the 20 of the 160 years in the perturbation to allow for equilibration. The years are matched between the control and perturbation such that we difference the same year. We have added the following text:

"We discard the first 20 years of each perturbation simulation (as with the mean surface temperature analysis), and use the corresponding matching years in the control run when creating the differences."

Page 6, lines 24-25: In light of Malavelle et al. (2018) (https://www.nature.com/articles/nature22974), it might be worth making a comment as to whether the cloud lifetime effect is actually something that should be included.

While the point about the uncertainty of the cloud lifetime effect and whether it should be included in models is well taken, the results of our paper are not authoritative on this subject, so any statement on the worthiness of inclusion of the cloud lifetime effect would be too speculative to be included in the paper.

Page 6, lines 24-29: Your results bear a striking similarity to those of Seneviratne et al. (2018): https://www.nature.com/articles/s41561-017-0057-5 Your choice, but could be worth a comment?

Yes, cited and added to the manuscript:
"The response pattern is also similar to regional modifications of land surface albedo as reported in Seneviratne et al. (2018)."

Page 10, lines 1-3: Is there a statistical test you could use? Anything better than simply eyeballing the change?

In lieu of another statistical test on the dataset (we note that the t-test for significance has already been applied), we have changed the text to be more descriptive and precise regarding shape:

"For the spatial average over the eastern US, the shape of the distributions remains unimodal and not skewed in GISS-E2 and GFDL-CM3, except for CESM1 which is not skewed in the control simulation but skewed in the perturbation. Global mean temperature distributions are consistently bimodal in the control and perturbation and generally not skewed. Overall, distribution shapes are mostly consistent, indicating that a mean shift is the statistical mechanism behind the increased temperature extremes."

Figure 1: The results here say some interesting things about aerosol transport. In particular, the bottom row indicates that Wang et al. (2014) (https://agupubs.onlinelibrary.wiley.com/doi/full/10.1002/2014JD022297) and subsequent related studies might be relevant. The point being, your discussion about common patterns of temperature change is good, but it's not necessarily the entire story.

Interesting point. We have added a citation to the Wang paper and the following sentence to the manuscript:
" The role of transport of BC from source regions remote to the Arctic may also be a contributor to the Arctic temperature response (Wang et al., 2014)"

Figure 2: I'd recommend getting rid of the red color and just leaving it white. It doesn't say much that two models agree on the sign but without statistical significance.

Done.

In Figure 4, CESM1 clearly has some different Southern Hemisphere behavior from the other two models (which doesn't show up in Figure 5, possibly due to cancellation). Any hypotheses as to what might be going on here?

There is indeed some different behavior, such as cooling over Australia, in CESM1 in the TXx response. The extreme temperature response is mostly not statistically significant in any simulation in CESM1 in the southern hemisphere, so this is most likely just internal variability.

**Reviewer 2**

Review of "Local and remote mean and extreme temperature response to regional aerosol emissions reductions" by Westervelt at al.

Summary
This paper quantifies the temperature response, including extremes, due to removal of aerosol emissions in six different geographical regions, with three different climate models. Such removal generally leads to warming, with the Arctic region particularly vulnerable. Similar changes occur in temperature extremes. The authors also update Regional Temperature Potentials, which are useful for estimating regional climate impacts due to regional aerosol changes in Integrated Assessment Models. Overall, the paper is well written and contains interesting and novel results. I recommend publication with minor revision.

We thank the reviewer for the useful review and comments.

Comments
P2 L12. See also "Emerging Asian aerosol patterns" Nature Geosci. 12 (2019) by Samset et al.

Good point. Added and cited.

There are also several PDRMIP papers that are relevant to the discussion, that should probably be cited. For example, "Rapid Adjustments Cause Weak Surface Temperature Response to Increased Black Carbon Concentrations" by Stjern et al. JGR 2017.

Added and cited. Refer to the comment below which also references this paper.

Also, "Arctic Amplification Response to Individual Climate Drivers" by Stjern et al. JGR 2019. Looks like the Arctic Amplification paper is eventually mentioned on Page 6.

Yes, this is already cited and discussed.

P4 L33. Naik reference parentheses typo.

Fixed, thanks.

Model Description Section. Please include some brief information of each of these model's aerosol ERF (e.g., PD-PI). For example, in "A 21st century northward tropical precipitation shift

caused by future anthropogenic aerosol reductions" by Allen JGR 2015, CESM yields - 1.52; GFDL yields -1.60; and GISS yields -0.76 W/m^2. So GISS has a much smaller aerosol ERF. Based on this alone, I would expect any temperature response to similar aerosol perturbations to be weakest in GISS. You may also want to mention why this is. For example, GISS lacks aerosol-cloud second indirect effects (aerosol-cloud lifetime effect). Looks like this is eventually mentioned on Page 6. Might be beneficial to mention here as well. Although, it is interesting that GISS also has a smaller climate sensitivity to CO2, relative to GFDL and NCAR. This could also contribute to a weaker surface temperature response in GISS.

Good points. We have added the following text:
"For comparison, the present day minus pre-industrial aerosol ERF in CESM1, GFDL-CM3, and GISS-E2 is -1.52, 1.60, and 0.76 W m$^{-2}$, respectively (Allen et al., 2015). This version of the GISS-E2 model does not include the aerosol-cloud lifetime effect, resulting in a smaller ERF, as discussed below."

The sensitivity to regional aerosol reductions in GISS-E2 is about the same or larger than GFDL-CM3 and CESM1 (Figure 8), suggesting that the weaker CO2 climate sensitivity is not a major factor in the GISS-E2 weaker temperature response.

Methods Section. These simulations are designed for an instantaneous removal of aerosol emissions. Thus, there is no time evolution of the aerosol emission reductions, which would be more realistic. How does this impact the results? Maybe a few comments on this are necessary.

These are idealized simulations, meant to establish the maximum potential impact of phasing-out of aerosols on climate. The temperature response would be a bit weaker overall assuming a ~100 year drawdown of aerosols. If only the years after the time evolving anthropogenic aerosol emissions approached 0 (or close to it) were included in the hypothetical analysis, then I expect the results to be similar to what we are showing here. We have added a line clarifying this in the manuscript:
"Aerosol emissions removals are instantaneous and we do not consider the effect of a long time-evolving drawdown."

Methods Section. It is stated perturbation simulations are conducted for 160-200 years. What years are actually analyzed? I assume the first several decades (hopefully more) are not used, since the model is not yet in some sort of quasi-equilibrium?

We discard the first 20 years of the perturbation simulation. This has been added to the manuscript. Also for the 400 year control simulation, the first 200 years are discarded as spin up. There is a tradeoff between years analyzed and likelihood that the model is in equilibrium. In order to generate robust statistics, we need a long sample size, so this is part of the reason why we only discard the first 20 years.

Added to manuscript:
"The first 20 years of the perturbation simulations are discarded in the response calculation."

P6 L26. "Surface temperature response is strongest in the US SO2 and Europe SO2 simulations in all three models, with annual mean local and remote temperature increases of up to 1 K or higher." I assume this is related to the magnitude of the aerosol emission reduction. What if you normalize by this perturbation? I guess you eventually normalize by ERF, which would be similar.

Yes, we normalize by effective radiative forcing, which provides the same effect (with different units). In a forthcoming paper in preparation, we plan to present some results normalized by emissions, so we leave that analysis for later.

P7 L10. Regarding the weak and inconsistent BC temperature response. Again, this appears to be consistent with the Stjern PDRMIP paper above. Also, this may be related the resulting rapid adjustments. See, for example, Smith, C. J. et al. "Understanding rapid adjustments to diverse forcing agents". Geophys. Res. Lett. 45, 12023–12031 (2018). And in particular, the impact of the vertical profile of absorbing aerosol on the rapid adjustments. For example, Allen, R. J. et al. "Observationally constrained aerosol cloud semi-direct effects" npj Climate and Atmospheric Science (2019) showed very different surface temperature responses to absorbing aerosol dependent on the vertical absorbing aerosol profile. Models tend to have a vertically-uniform profile, and this leads to a negative adjustment, associated with high cloud reductions. However, a vertical profile that resembles CALIPSO observations (more absorption in the lower troposphere) yields low/mid-level cloud reductions, a positive adjustment, and surface warming. Looks as if this is eventually discussed near P11.

Yes, the Smith et al. (2018) paper is already cited and discussed on page 11. However, we have added another reference to it, as indicated below. We agree with the reviewer that rapid adjustments could explain some of the weaker responses for BC reductions. We have added the following to the text:

"The weak forcing in the black carbon simulations may reflect the role of rapid adjustments (Stjern et al., 2017; Smith et al., 2018), including the semi-direct effect of BC on clouds (Allen et al., 2019)."

Figure 2. Can the "Robustness %" be defined in the figure caption.

Yes, done. Added to caption:
"Robustness indicates percentage of the surface area that has all 3 models in sign agreement."

Near Page 8 L10. Can you explain why BC reductions lead to surface warming (despite having a positive RF) in nearly all cases?

This is likely a symptom of internal variability swamping the signal from BC reductions. For example, BC reductions in India yield very little statistical significance in temperature response.

How do tropical emission reductions drive Arctic warming? Is it due to direct transport of aerosol to the Arctic, or due to changes in atmospheric/oceanic circulation that then leads to

Arctic warming? Are these responses robust across the three models? Perhaps this is outside the scope of the current study.

We believe that the majority of the impact is due to atmospheric and ocean circulation that then causes Arctic warming. The first reviewer pointed out that transport of BC to the Arctic is possible even from the tropics, though Wang et al. (2014 suggests that tropical regions such as Southeast Asia seem to contribute less than 10% to the BC burden in the Arctic, so we do not expect this to be the dominant pathway for Arctic temperature changes. Figure 3b shows the Arctic temperature response, and for some of the tropical aerosol emissions pertubations (such as India SO2, Africa BB, etc), there is some minor degree of robustness. For example, for India SO2 and South America BB, all three models agree on sign.